

# Spatiotemporal patterns of urban thermal environment and comfort across 180 cities in summer under China's rapid urbanization

Zhibin Ren[1], Yao Fu[1], Yunxia Du[1] and Hongbo Zhao[2]

[1] Key Laboratory of Wetland Ecology and Environment, Northeast Institute of Geography and Agroecology, Chinese Academy of Sciences, Changchun, China
[2] Key Research Institute of Yellow River Civilization and Sustainable Development & Collaborative Innovation Center on Yellow River Civilization of Henan Province, Henan University, Kaifeng, China

## ABSTRACT

**Background**. China is considered as the largest and most rapidly urbanizing nation in the world. However, possible changes of urban thermal environment and comfort under the rapid urbanization in China still remain poorly understood at a national scale.

**Methods**. Based on the data collected from 180 cities in 1990, 2005, and 2015 in China, the spatiotemporal patterns of urban thermal environment and comfort in summer and their relationships with urbanization variables were investigated in this study.

**Results**. Our results indicate that urban thermal environment has changed greatly during the 25 years. Furthermore, the changes of urban climate in different regions are inconsistent. The Physiological Equivalent Temperature (PET) at most cities (81%) in China increased from 1990 to 2015, which suggested that urban thermal comfort in China was also deteriorating during the 25 years. However, while the PET of some cities in China began to decrease from 2005 to 2015, there were still 33% of cities that had positive trends,which mainly located in North region. Urbanization resulted in a significant influence on urban climate. Compared to southern cities, northern cities were more sensitive to urbanization impact. The most important contribution to increasing of PET for urbanization variables is gross domestic product, followed by urban population. The analysis results reveal changing patterns of urban thermal comfort in China during summer season. It can help urban government and managers improve urban thermal environment and comfort.

## INTRODUCTION

Many parts in the world have experienced the rapid urbanization during last few decades (*Montgomery, 2017*; *Szabo, 2018*). The expansion of urban impervious surfaces can especially change urban thermal environment, which makes significant impacts on urban life quality and even urban sustainable development (*Cao et al., 2018*; *Lin et al., 2018a*; *Lin et al., 2018b*). The high urban temperature and intensive heat can result in

Corresponding authors
Yunxia Du, duyunxia@iga.ac.cn
Hongbo Zhao,
10340024@vip.henu.edu.cn,
zhaohbhhwm@163.com

high cooling energy consumption (*Giridharan & Emmanuel, 2018*; *Kolokotroni, Zhang & Watkins, 2007*), raise pollution levels (*Sarrat et al., 2006*; *Weng & Yang, 2006*), reduce urban thermal comfort (UTC) (*Basu & Samet, 2002*; *Klenk, Becker & Rapp, 2010*; *Salata et al., 2018*), decrease outdoor activities and cause serious public health problems (*Wong et al., 2018*; *Sagris & Sepp, 2017*). The urban thermal environment may be expected to become worse in the coming years with two significant escalating global trends (urban heat islands(UHI) and global warming). Therefore, a thorough understanding of urban thermal environmental patterns is critical for urban planners to adapt effective mitigation measures and strategies to improve urban thermal environment, especially in summer time.

Urban thermal environmental issues have recognized by scientists and urban planners during the past several decades. Previous studies on the urban thermal environment have relied on land surface temperature extracted from remote sensing (e.g., *Weng, Liu & Lu, 2007*; *Cao et al., 2010*; *Lin et al., 2018a*; *Lin et al., 2018b*) and air temperature observed at weather station (e.g., *Oxoli et al., 2018*; *Wang et al., 2018*). The urban temperature has been the most intuitive performance indicator for analyzing the urban thermal environment (*Lan & Zhan, 2017*; *Moustris et al., 2018*). However, the urban air temperature is an inadequate measure to investigate urban thermal environment (*Aljawabra & Nikolopoulou, 2018*; *Cheung & Jim, 2018*). Moreover, urban thermal comfort is a state of mind that articulates satisfaction with a specific urban thermal environment, depending on a combined effect of the physical and climatic parameters (*Aljawabra & Nikolopoulou, 2018*; *Galagoda et al., 2018*). When analyzing the urban thermal environment, it is important to bear in mind that the other factors could also influence urban thermal environment such as solar radiation, humidity, and wind speed besides the air temperature (*Lee & Mayer, 2018*; *Yang, Lin & Li, 2018*). Therefore, the urban thermal comfort can better render the implication of the urban thermal environment because it considers an integral effect of climatic parameters, such as, air temperature, wind speed, air humidity and solar radiation (*Salata et al., 2017*). When urban residents are under more thermal stress, an urban accompanying discomfort could negatively affect their outdoor life and health in many ways (*Salata et al., 2017*; *Cao et al., 2018*). Therefore, a better understanding and monitoring of the urban thermal comfort is necessary and important for developing strategies for urban planners, and policy makers to mitigate urban thermal stress and improve urban life quality in cities.

China as a developing country has experienced a rapid urbanization in the past six decades. In China, some studies showed that urbanization has contributed greatly to the observed warming since the 1960s (*Tarabon et al., 2018*; *Wang et al., 2019*). By 2030, urban dwellers in China will increase by nearly 200 million (*Wu, 2014*), which will further exacerbate urban warming stress on humans. However, how the urban thermal comfort changes in China under the rapid urbanization is not well understood yet. Furthermore, most previous efforts about urban thermal comfort just focused on several selected measurements by using conventional meteorological monitoring approaches in a single city or a few big cities (e.g., *Linden, Fonti & Esper, 2016*; *Cheung & Jim, 2018*; *Xu et al., 2018*). This may not only understand the thermal comfort in single cities, but also fail to reveal the detailed patterns of urban thermal comfort at a national scale (*Huang et al., 2016*; *Zhou et al., 2018*). Spatiotemporal patterns of urban thermal comfort under the rapid

urbanization in China are still poorly understood at the national scale. Therefore, more detailed studies on nationwide cities and across different climatic zones are needed. The better understanding of spatiotemporal change of urban thermal comfort is imperative today especially when China is constrained by both urbanization and global climate change.

In this study, based on long-term daily meteorological observations and urbanization information collected from the Statistics Yearbook of Cities in 1990, 2005, to 2015 in 180 cities of China, we propose to analyze spatiotemporal patterns of urban thermal environment and comfort in the cities. More specific research objectives include: (1) to investigate spatiotemporal changes of urban climate under a rapid urbanization in 1990, 2005 and 2015, (2) to explore the spatiotemporal changes of urban thermal comfort in the 180 cities in summer under the rapid urbanization, and (3) to analyze relationships between thermal comfort index and urbanization factors. The expected research results would be helpful for better understanding of relationships between urbanization and urban thermal comfort and provide useful information to urban government and planner to improve urban thermal environment.

## METHODS

### Study areas

In our study, the 180 cities under rapidly urbanizing process were selected due to the availability of relevant meteorological and urbanization data in China (Fig. 1). The administrative boundary of cities in China often includes very large areas of countryside. In this study, only the built-up area of each city was analyzed, without considering the whole area within the administrative boundary. The built-up area refers to the urbanized area of a city (National Bureau of Statistics of China, 1990–2010). In order to explore regional patterns in urban thermal environment, the selected cities in our study were then divided into seven different regions (Northeast, North, South, Northwest, Southwest, Central and East), according to regional division criterion used by the national government (*Zhao et al., 2013*). These seven regions include all the provinces of mainland China.

### Data collection
#### *Meteorological data*

Given the long-term meteorological observation data from 1980 to 2016, we found that there were not extreme climatic events occurring in years 1990, 2005 and 2015 and the climate in 1990, 2005, and 2015 could almost represent the average climatic condition of 1990s, 2000s, and 2010s (Table 1). Therefore, the historical publicly available meteorological data observed in 1990, 2005 and 2015 from the National Meteorological Information Center (http://data.cma.cn/) were used in our study in order to investigate spatiotemporal trends of urban thermal comfort. The monthly meteorological parameters observed in summer months (June, July and August) used in this study included monthly average values of air temperature, relative humidity, wind velocity and global radiation. For every city, all meteorological parameters observed from all the Chinese surface meteorological stations in urban built-up area were used in our study and such kind of data has been widely used to study the urban thermal environment (e.g., *Han, Tang & Xu, 2019*; *Liu et al., 2019*; *Yang,*

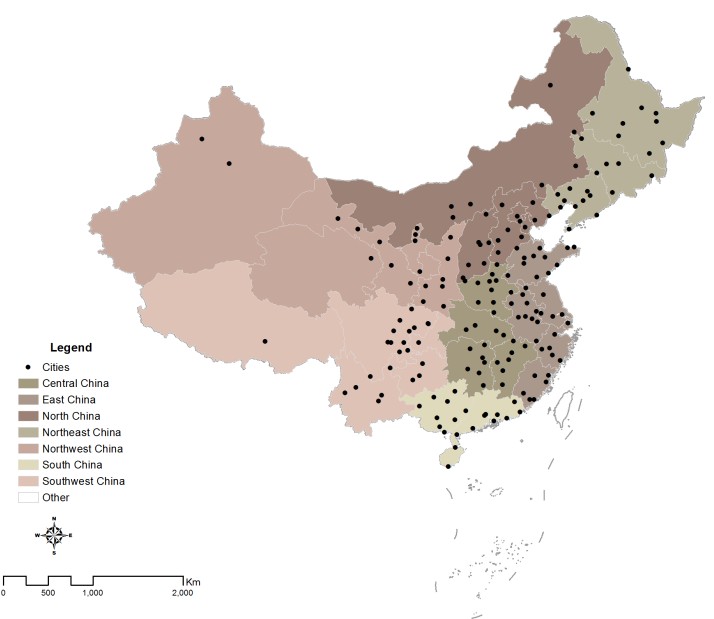

**Figure 1** Spatial distribution of 180 cities in China.

**Table 1** The average urban climte in 1990, 2005, 2015, 1990s, 2000s, and 2010s in China.

| Urban climate | 1990 | 2005 | 2015 | 1990s | 2000s | 2010s |
|---|---|---|---|---|---|---|
| Temperature (°C) | 25.3 | 26.8 | 26.1 | 24.9 | 27.1 | 25.8 |
| Humidity (%) | 75.8 | 70.4 | 68.3 | 75.13 | 71.3 | 67.7 |
| Wind speed (m/s) | 3.1 | 2.1 | 2.3 | 2.8 | 1.9 | 2 |
| Solar radia (W/m$^2$) | 393.4 | 389.5 | 401.2 | 390.5 | 386.7 | 398.5 |

*Huang & Tang, 2019*). The four variables from the meteorological stations in June, July and August were also averaged into the summer season. Our focus is on the summer season because hundreds of millions of urban residents in China endure their highest heat-related stress during the time period of a year.

### Urbanizaiton data

All the data of urbanization variables were collected from the Statistics Yearbook of Cities in China (National Bureau of Statistics of China, 1990–2016). This series of yearbooks has been compiled and published annually by the National Bureau of Statistics since 1985. The document data published in each yearbook in a given year were from the previous year. The yearbooks published between 1991 and 2016 were used to collect data for the 25 years between 1990 and 2015. Urbanization is a process marked by urban area expansion with intensive land use change, economic development, and rapid population growth (*Weber & Puissant, 2003*). Therefore, all the continuously recorded variables related to population, land cover and economic development were used to analyze urbanization impact on urban thermal environment in our study. There were six variables that were defined or calculated according to the official statistical yearbooks (National Bureau of Statistics of China,

1991–2016), including total urban population, urban population in build-up area, built-up area (km$^2$), population density (n/km$^2$), GDP ($10^8$ Yuan), per capita GDP (Yuan/person) and urban vegetation coverage (%).

## Calculation of Urban thermal comfort

Physiological Equivalent Temperature (PET) is one of the most regularly used thermal comfort index to assess the human thermal condition, which has been widely applied in different climatic zones (*Potchter et al., 2018*). PET is based on human energybalance in terms of the Munich Energy Balance Model for individuals (*Matzarakis, Rammelberg & Junk, 2013*). It is a universal index which integrates the thermo-physiology of the human body (sex, height, activity, clothing resistance for heat transfer,shortwave albedo and long wave emissivity of the surface) and the multiple relevant meteorological parameters including air temperature, relative humid, wind velocity etc. PET is derived from the human energy balance and expressed in the unit (°C) as an indicator of thermal stress, which makes the result more comprehensible for urban and regional planners, decision-makers, and even the public (*Froehlich & Matzarakis, 2018*). Based on the monthly average values of air temperature, relative humidity, wind velocity and global radiation for summer in 1990, 2005, and 2015 in this study, the monthly PET index for urban stations for every city in June, July and August was calculated by RayMan software (*Matzarakis, Ivanova & Balafoutis, 2007*; *Matzarakis & Endler, 2010*). The thermos-physiological parameters of the human beings used in this research were set up as a typical male 35 years old, 1.75 m height, 75 kg weight, with an internal heat production of 80 W and a heat transfer resistance of the clothing of 0.9 clo. The assessment scale used in this study to classify cold stress (PET <13 °C), thermal comfort (13 $\leq$ PET $\leq$ 29) and heat stress (PET > 35 °C) is described in Table 2.

## Data analysis

The values of the PET in June, July and August for urban stations in each city were first averaged into a summer PET for all following analyse. Based on the values of summer PET in 1990, 2005, and 2015, spatial change patterns of PET among the three years in different regions in China were then conducted in our study.

By analyzing the distribution of urbanization variables, we found that that GDP, per capita GDP, built-up area, total population, population density, population in the built-up area had a highly skewed distribution. Therefore, these variables were in-transformed before correlation and regression analyse in order to improve the comparability and the linearity of relationships between PET and a set of urbanization variables. In order to analyze the impact of urbanization on PET, the research on the relationship between PET and urbanization variables was conducted. First, the Pearson correlation analysis and correlation coefficient were conducted between these independent and dependent variables. Second, a simple linear regression model was used to determine the effect of urbanization on PET. In our analysis, these six urbanization variables in the selected three years (1990, 2005, and 2015) were used as independent variables and the corresponding average PET was used as a dependent variable. As most variables were significantly correlated with each

**Table 2  Thermal sensation classes for human beings.**

| PET (°C) | Thermal perception | Grade of physical stress |
|----------|--------------------|--------------------------|
| >42 | Very hot | Extreme heat stress |
| 35–42 | Hot | Strong heat stress |
| 29–35 | Warm | Moderate heat stress |
| 23–29 | Slightly warm | Slight heat stress |
| 18–23 | Comfortable | No thermal stress |
| 13–18 | Slightly cool | Slight cold stress |

**Table 3  The statistical description of urbanization and meteorological parameters in China.**

| Urbanization and urban climate | 1990 | 2005 | 2015 |
|--------------------------------|------|------|------|
| urban vegetation cover (%) | 17.9a | 32.2b | 38.8c |
| Urban population ($10^4$) | 58.4a | 118.9b | 136.6c |
| Built-up area ($km^2$) | 45.4a | 117.9b | 160.2c |
| Population density ($n/km^2$) | 12013.3c | 9629.8b | 7777.9a |
| GDP | 38.3a | 785.7b | 1583.9c |
| Per capita GDP | 5806.9a | 53105.7b | 68682.6c |
| Temperature (°) | 25.3a | 26.8b | 26.1b |
| Humidity (%) | 75.8a | 70.4b | 68.3c |
| Wind speed (m/s) | 3.1a | 2.1b | 2.3b |
| Solar radia ($W/m^2$) | 393.4a | 389.5a | 401.2a |

**Notes.**
Values with different letters indicate statistically significant differences among the three different years at $P$-value <0.05.

other, we finally conducted a multiple regression analysis in which PET was the response variable and urbanization variables were predictors. All statistical analyses were carried out by using SPSS19.0.

# RESULTS

## Spatiotemporal changes of urbanization and meteorological parameters in China

Per the seven variables of urbanization in China, the statistical results showed that China experienced a rapid urbanization during the last 25 years (1990–2015) (Table 3). Compared to 45.4 $km^2$ in 1990, the mean urban build-up area reached to 117.9 $km^2$ and 160.2 $km^2$ in 2005 and 2015 respectively. Urban population and GDP increased greatly from 1990, 2005 to 2015. During the last 25 years, urban vegetation coverage also increased sharply (Table 3).

Due to the rapid progress of urbanization such as land cover change and the increasement of urban population over China, urban thermal parameters, such as, air temperature, humidity, wind speed, solar radiation parameters could correspondingly be changed correspondingly. The average air temperature increased significantly from 1990 to 2005 and decreased slightly from 2005 to 2015 (Table 3). Analysis of the average value of air temperature in different regions (Fig. 2) indicated that different changes have occurred
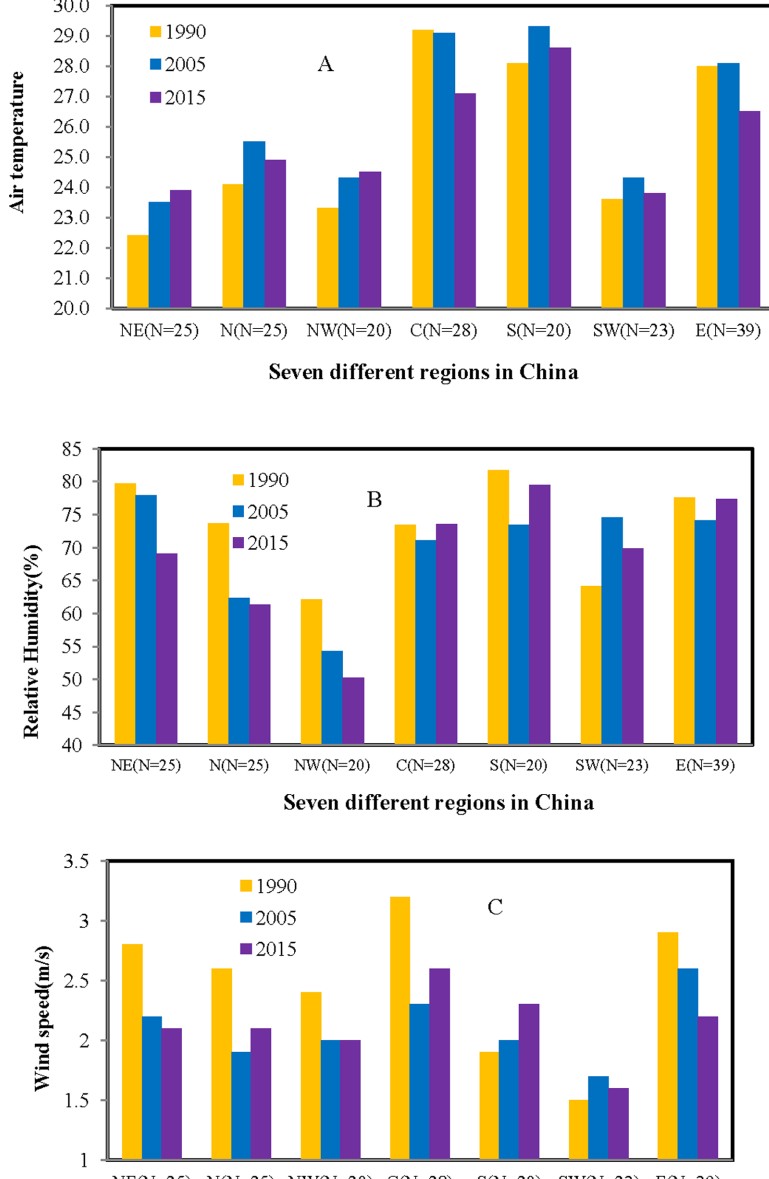

**Figure 2** Changes of urban thermal parameters across different regions in China (NE, Northeast; N, North; NW, Northwest; C, Central; S, South; SW, Southwest and E, East)((A) Air temperature, (B) Relative humidity, (C) Wind speed).

across different regions from 1990 to 2015. The air temperature in Northeast and Northwest increased gradually from 1900, 2005 to 2015. However, it decreased from 1990 to 2015 in Central and East China. The relative humidity and wind speed showed significant decreasing changes, especially in Northeast, North, and Northwest regions.
**Table 4  Descriptive statistics of PET for 180 cities in China.**

| Year | 1990 | 2005 | 2015 |
|---|---|---|---|
| Min | 13.2 °C | 17.5 °C | 14.7 °C |
| Max | 34.3 °C | 37 °C | 39.3 °C |
| Average | 28.7 °C | 31.7 °C | 30.6 °C |
| SD | 3.9 °C | 3.4 °C | 3.3 °C |

## Spatiotemporal changes of PET across 180 cities in hot summer

The average PET increased significantly from 1990 to 2015 (Table 4), which means people suffer from more and more heat stress during the summer period and feel extremely uncomfortable. For the 180 cities in 1990, the PET ranged from 13.2 °C to 34.3 °C with an average of 28.7 °C. The urban average PET reached up to 31.7 °C with an increase rate of 11% from 1990 to 2005. Compared to the average PET in 2005, it slightly decreased in 2015 with the average value of 30.6 °C. Analysis of the average value of PET in all different regions in mainland China showed that cities in the South region had the highest level of PET with 33.5 °C, followed by the Central region. The lowest levels of PET have occurred in Northeast and Northwest region with only 27 °C (Fig. 3). The frequency analysis of PET in 180 cities (Fig. 4) indicated showed that the highest frequency for PET between 29 and 35 was 60% (Moderate heat stress) in 1990 and it increased gradually to 63% in 2005, and 70% in 2015. Meanwhile, the frequency for PET between 18 and 23 (No thermal stress) decreased gradually from 8% in 1990 to 0.55% in 2015.

From 1990 to 2005, the PET of most cities in China exhibited an increasing change (Figs. 5–6). Of the 180 cities investigated, ninety-five percent of them had positive change values, ranging from 0.1 °C to 6.8 °C (Figs. 5–6) and 80% of the values were larger than 1.5 °C, with a highest rate of increase occurring in North region (4 °C), followed by Northeast region(3.4 °C). From 2005 to 2015, the PET in some cities in China began to exhibit a decreasing change (Fig. 6). Sixty-seven percent had negative change values, especially in South region. However, there are still 33% of cities had positive values, mainly located in North region (Figs. 6–7). Overall, the PET of most cities (81%) in China exhibited an increasing change (Figs. 6–7) from 1990 to 2015.

## Effect of urbanization on PET in hot summer in China

Table 5 lists the Pearson's correlations between PET and urbanization variables. Five of the seven urbanization variables had significant positive effects on PET except for population density. Meanwhile, urban vegetation coverage had a negative relationship with PET. In addition, the increase 1 in Ln(GDP) resulted in an increase of 0.67 for PET (Fig. 7). The increase 1 in Ln(total population) led to an increase of 0.61 for PET. From the $R^2$ value in Fig. 7, we can easily see that the variables of GDP and total population could explain 22.0% and 13.2% of variance of PET, respectively. Finally the multiple regression analysis showed a constructed model using the four most significant variables as follows:

$$y = -25.759 + \_1.256b_1 - 2.036b_2 + 1.104b_3 + 0.051b_4$$
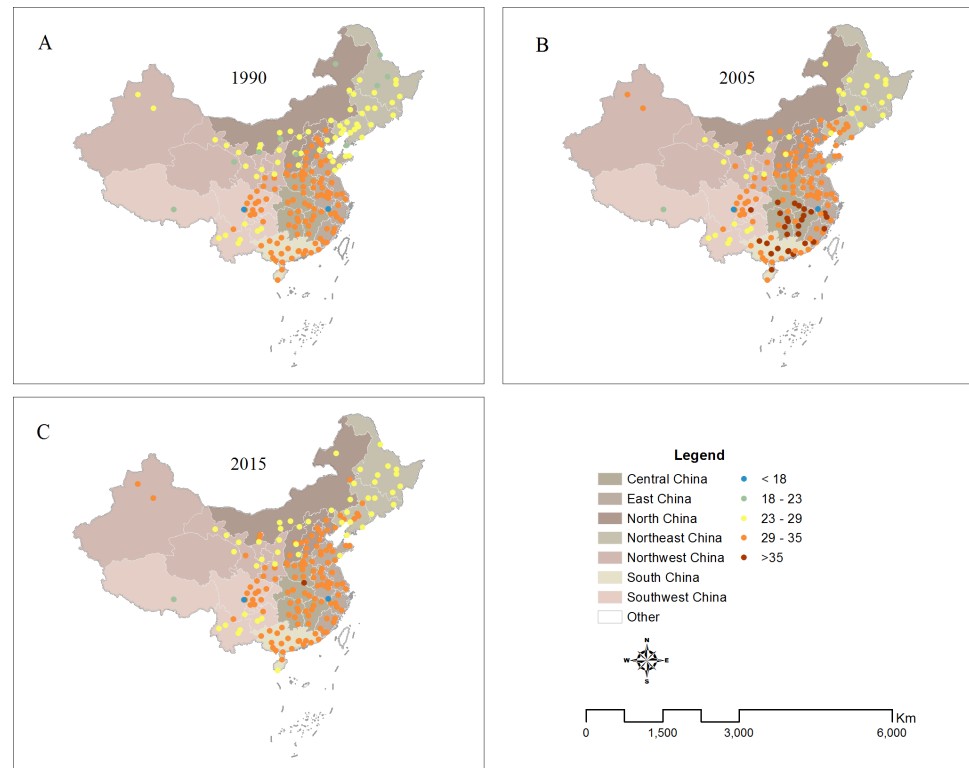

**Figure 3** Spatial pattern of PET across different regions in China ((A) PET in 1990, (B) PET in 2005, (C) PET in 2015).

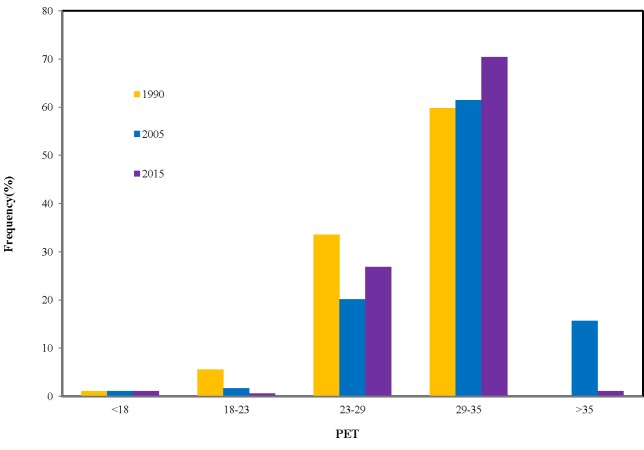

**Figure 4** Histogram of the frequency statistics for PET.

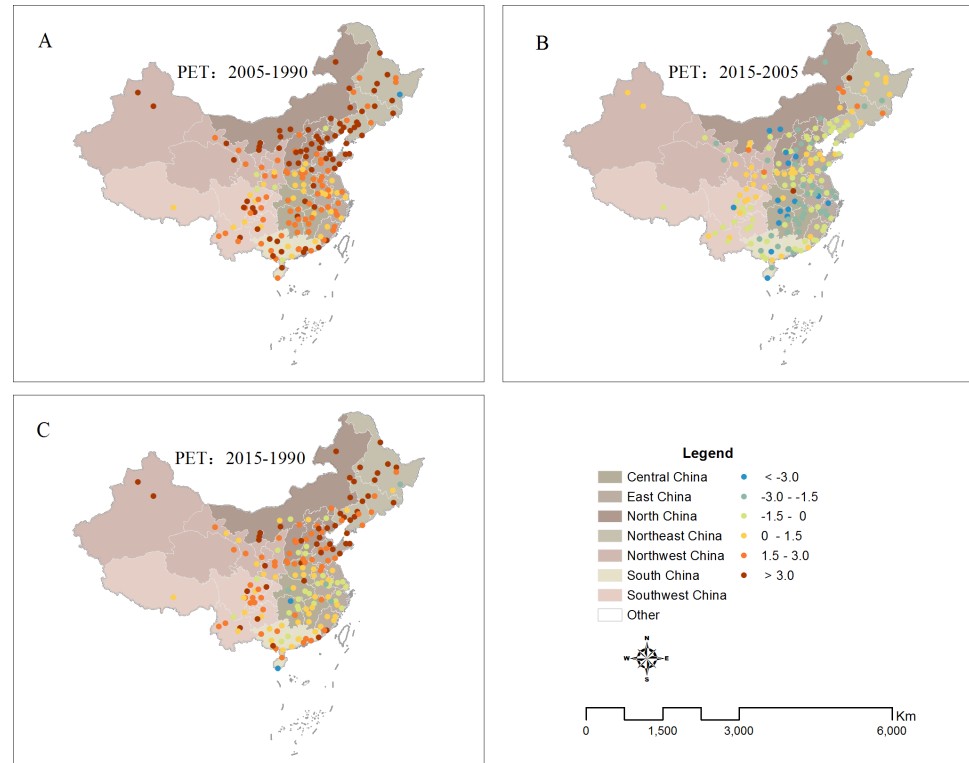

**Figure 5** Spatial patterns of PET changes in different time periods and regions in China ((A) PET from 1990 to 2005, (B) PET from 2005 to 2015, (C) PET from 1990 to 2015).

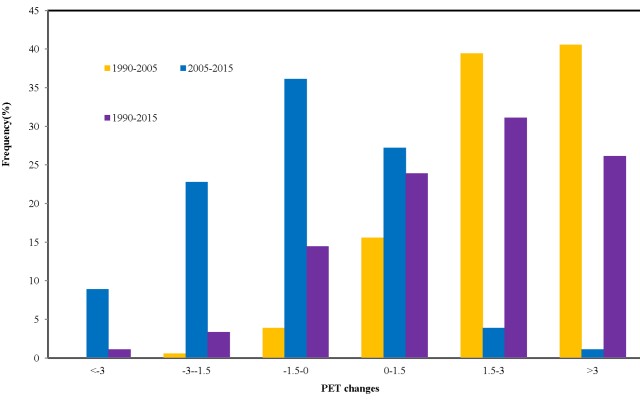

**Figure 6** Histogram of the frequency statistics for PET changes across different years.

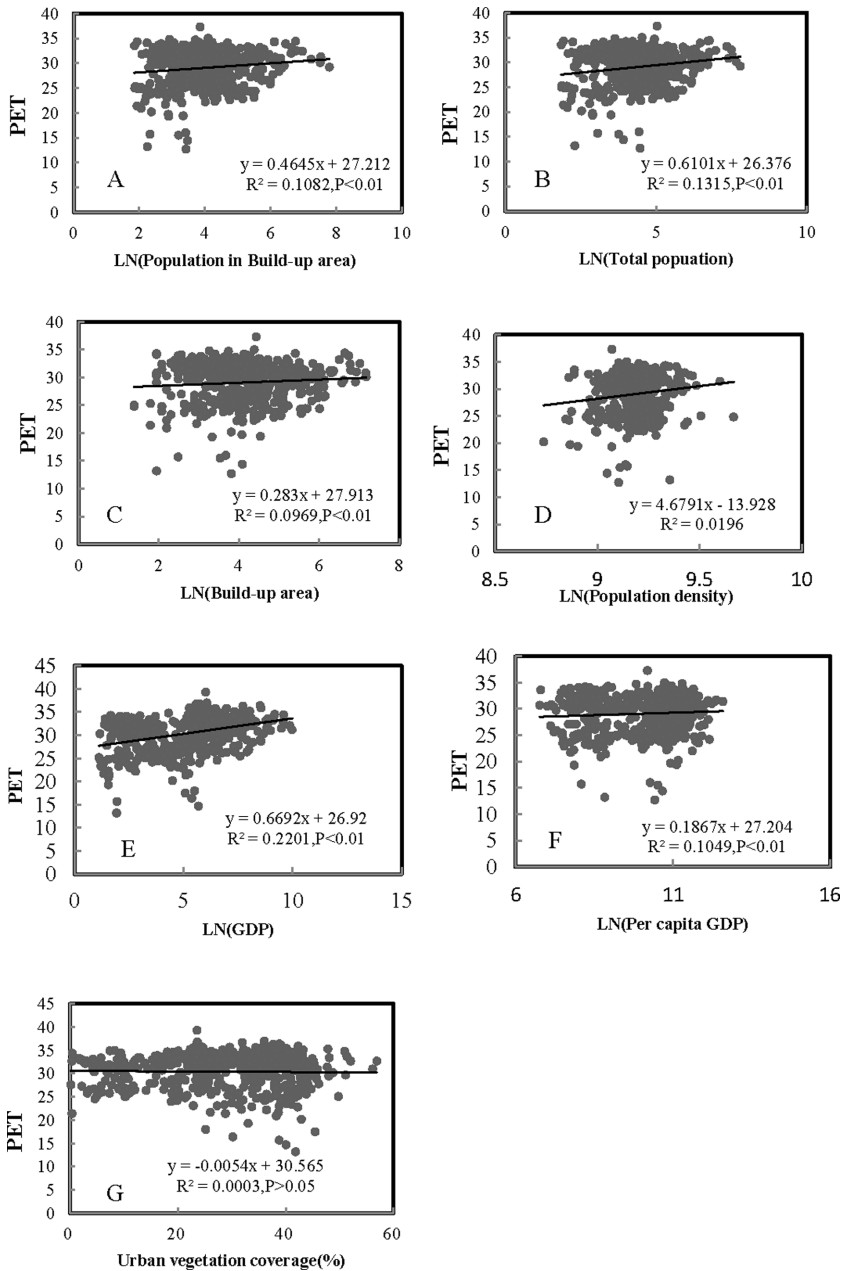

**Figure 7** Regression analyses of urbanization with PET ($n = 540$) ((A) Population in build-up area, (B) total population, (C) build-up area, (D) population density, (E) GDP, (F) per capita GDP, (G) urban vegetation coverage).

**Table 5  Pearson correlation coefficients between PET and urbanization ($n = 540$).**

| Urbanizaiton indices | Pearson coefficients |
|---|---|
| Total urban population | 0.136[**] |
| Urban population in build-up area | 0.182[**] |
| Built-up area (km2) | 0.164[**] |
| Population density (n/km2) | 0.022 |
| GDP (108 Yuan) | 0.168[**] |
| Per capita GDP (Yuan/person) | 0.234[**] |
| Urban vegetation coverage (%). | −0.018 |

Notes.
[**]Correlation is statistically significant at the 0.01 level (two-tailed).

**Table 6  Summary of the multiple regression models, in which PET is a response variable, and urbanization variables are predictors.**

| Variable | Lable | Cofficient | Std Coeffcient | $t$ | $p$ |
|---|---|---|---|---|---|
| Constant | | 25.759 | 0 | 32.667 | <0.001 |
| GDP | b1 | 1.265 | 0.655 | 8.443 | <0.001 |
| BA | b2 | −2.039 | −0.573 | −6.072 | <0.001 |
| TP | b3 | 1.104 | 0.307 | 3.842 | <0.001 |
| VS | b4 | 0.051 | 0.167 | 3.822 | <0.001 |

Notes.
BA, built-up area (km$^2$); TP, total urban population; VC, urban vegetation coverage (%).

where $y$ is the green space coverage and $b$1-4 are the variables shown in Table 6. This model explained 31.6.7% of the variance of PET.

## DISCUSSION

China has experienced the rapid urbanization during the last 30 years (*Jiang et al., 2019*; *Lu et al., 2019*), which could definitely cause changes of urban climate. The warming change caused by urbanization has already been well-documented (e.g., *Lin et al., 2018a*; *Lin et al., 2018b*; *Fu & Weng, 2018*; *Yang, Huang & Tang, 2019*). Our results have also agreed to those existing studies , which showed that the average air temperature increased significantly from 1990 to 2005 and to 2015. Such increasing changes may be linked to appearances of increasing impervious surfaces during the rapid urbanization progress. The greater absorption capacity of solar radiation, heat capacity and conductivity in the urban area, resulted in more heat storage (*Zhang & Sun, 2019*), which leads to a substantial increase in sensible heat flux during daytime and heat release at night. Our studies also demonstrated showed that the relative humidity appeared the significant decreasing changes, which might be caused by huge losses of leaf area index, green vegetation fraction, and so on because of urban sprawl and land use and land cover change) (LULC) (*Hernandez-Moreno & Reyes-Paecke, 2018*; *Sun et al., 2018*). In addition,the mean wind speed for 180 cities also showed significant decreasing changes from 1990 to 2005, and to 2015. The surface roughness over urban region was significantly altered because of the emergence of high-rise

buildings and other LULCC (*Walker, 2011*; *Anup, Whale & Urmee, 2019*), which caused surface drag force increases and then decreased the wind speed in cities.

In recent years, many studies have only used single-climate parameters, such as, air temperature (Ta), to serve as indicators of urban thermal environment (*Weng, Liu & Lu, 2007*; *Oxoli et al., 2018*). In fact, the temperature factor only could not represent adequately urban thermal environment. Urban thermal comfort could better represent urban thermal environment because of consideration of the integral effect of the climatic parameters of the air temperature, wind speed, air humidity and solar radiation (*Salata et al., 2017*). Our results indicated that the PET of most cities (81%) in China exhibited a great increasing change (Figs. 3–4) from 1990 to 2015. The impact of urbanization on temperature and thermal comfort is quite different among the different cities and regions in China. Our research also indicated that the PET changes were much larger than air temperature changes associated with urban thermal conditions. This is because that urbanization has a negative impact on relative humid and wind velocity in urban area in addition to the warming change. Therefore, thermal comfort should be a better indicators to characterize the urban thermal environment than the urban air temperature.

In addition, our results also showed that cities in the South region had the highest level of PET (33.5), followed by the Central region and Northeast and Northwest region with only 27 °C (Fig. 3), which could be closely linked with the different regional climates. Many studies found that the higher temperature and humidity over China during summer is mainly distributed in the southeast of China, and relatively lower temperature and humidity concentrates in the western and northern of China, which may cause the different regional patterns of PET (e.g., *Li & Zha, 2018*; *Yao et al., 2017*; *Peng et al., 2018*). Our results furtherly indicated that the change of PET also exhibits a regional characteristic, suggesting larger increasing trends occurring in Northeast and North China from 1990 to 2005, and to 2015 and some decreasing trends in South regional cities from 2005 to 2015 (Figs. 5–6). There may be several reasons for the spatial inconsistent changes of urban thermal comfort in these cities, which maybe closely related with the different processes of urbanization in different regions in China. The economic development in the north is dominated by heavy industries characterized with high energy consumption (*Zhang & Sun, 2019*; *Pan et al., 2019*). The average summer temperatures in Northeast and North regions could be increased greatly by waste heat released by secondary industry. Therefore, PET in the cold northeastern China is more susceptible to the impact of urbanization. However, the economic development in the south is dominated by light industries characterized with high technology (*Wu et al., 2019*; *Liu et al., 2019*). Many cities in South region have higher environmental efficiencies than those northeastern cities. Some regulations of reforming technology or other measures like recycling wastes intensively and subsidizing the sewage disposal plants in recent years play an important role in reducing emissions (*Ma, Cai & Wu, 2019*). Therefore, the PET increased slower in north regions from 1990 to 2005 and even decreased from 2005 to 2015 than that in south regions. It might also be because that China has put ecological civilization as a national strategy to build a beautiful China in recent years (*Shi et al., 2019*; *Geall & Ely, 2018*). This strategy became a great opportunity for urban forest development in China. The government strengthens the building of

China's urban forest. Urban forest amounts and quality also have been further improved, especially for south regions ,which has improved urban thermal environment to a certain extent (*Liu et al., 2018*).

Most of the previous studies analyzed relationships between urban air temperature and its drivers across cities over time (*Lin et al., 2018a*; *Lin et al., 2018b*; *Fu & Weng, 2018*; *Yang, Huang & Tang, 2019*). The correlation analyses between PET and associated drivers can be conducted across space and time (*Yang, Huang & Tang, 2019*; *Cao et al., 2018*). In this study, we performed correlation analyses across both cities and years, and a series of new and different findings were revealed. In this study, increased GDP and population might be an important reason for the increased PET (Tables 4–5). The increasing GDP and population due to the urbanization might result in decreasing the vegetation coverage, and increasing amounts of roads and buildings and energy consumption (*Hernandez-Moreno & Reyes-Paecke, 2018*; *Sun et al., 2019a*; *Sun et al., 2019b*), and thus increasing the PET indirectly. The growing concentration of GDP and population is the most basic connotation of urbanization, The secondary sector of the economy for GDP includes that industries consume large quantities of energy and require factories and machinery to convert the raw materials into goods and products and produce waste heat that may cause thermal environmental problems (*Wu et al., 2019*; *Zhang & Sun, 2019*). Besides, the heat generated by human body and people's daily life, such as, air conditions and cars, has also caused the increasing artificial heat emissions (*Liu et al., 2018*; *Ma, Cai & Wu, 2019*; and thereby affected people's thermal comfort. Our results indicated that the correlation between urban vegetation coverage and PET is not significant. However, many previous studies demonstrated that vegetation coverage has a remarkable impact on regional climate (e.g., *Zhao & Wu, 2014*; *Alo & Wang, 2010*; *Chen, Ma & Zhao, 2017*). It might be because that urban vegetation was scattered and heterogeneous, surrounded by many impervious surfaces, which could reduce the ecological function of vegetation, such as, decreasing temperature and increasing humidity, etc (*Ren, He & Pu, 2018*; *Wang et al., 2019*). Besides, urban vegetation amount is relatively small compared with natural vegetation.

Undeniably, there exist some limitations for our research. It should be noted that PET intensity was obtained by only three years (1990, 2005, and 2015) for summer in our research. The PET in more continuous years should be used to conduct the spatiotemporal change patterns of urban thermal environment.In addition, PET might not change significantly in cities over years, especially when averaged into a single season Our study did not capture the severity of urban thermal discomfort and neither its peaks, and the averaged PET for a summer season might only provide a general condition for each city in each year. Therefore, more temporal studies should be conducted to explore the daily variation of PET in the future to reveal some more detailed information about PET (*Ren, He & Pu, 2018*). Besides, the correlation analyses conducted in different regions may lead to different results according to previous studies (*Lin et al., 2018a*; *Lin et al., 2018b*; *Cao et al., 2018*; *Sun et al., 2019a*; *Sun et al., 2019b*). Therefore, there are some limitations existing in the correlation analyses across different regions. For comprehensive understanding relationships between PET and drivers, it is necessary to analyze the relationships across different regions. However, if we would like to accurately identify the associated drivers and

mitigation strategies of PET in single city, it is better to analyze the relationships between PET and its drivers across many continuous years.

Given the general circumstance of global climate warming, the frequency and intensity of urban heat events would increase, especially in developing countries such as China (*Sun et al., 2019a*; *Sun et al., 2019b*) and urban thermal environment could be deteriorated in the future. Our study presented here will have some great management implications with many benefits to the urban planning, healthcare organizations and the power sector. Existing research has showed that the increase of urban air temperature could result in more usage of electricity (*Akbari, Pomerantz & Taha, 2001*; *Limones-Rodriguez et al., 2018*). Our study results could provide the basic information associated with the urban thermal environment to city power agencies. The spatiotemporal patterns of summer PET intensity could help the power agencies get the power demand variability at different cities. In addtion, many cities are aggressively taking some measures such as cool/green roofs and urban greening to reduce the urban heat island phenomena and moderate urban thermal environment (e.g., *Akbari, Konopacki, & Pomerantz, 1999*; *Lai et al., 2019*). Information on monthly and seasonal changes of urban thermal comfort could help these governmental agencies quantify the potential benefits of various adaptation strategies.

## CONCLUSIONS

Rapid urbanization has a great impact on urban thermal environment. In this study, we conducted the analysis of changing patterns of urban thermal environment and comfort for 180 cities in China over the past 25 years and attempted to explore relationships between urbanization variables and urban thermal comfort. By this study, several conclusions may be derived from our experimental results as follows:

(1) Urban thermal environment and comfort has changed greatly during the 25 years and the changes of urban climate in different regions are inconsistent.

(2) From 1990 to 2015, the PET of most cities (81%) in China exhibited an increasing change, especially in North region (4 °C), followed by Northeast region(3.4 °C). From 2005 to 2015, the PET in some cities in China began to exhibit decreasing changes. However, there are still 33% of cities that had positive trends, mostly occurring in North region in China.

(3) Urbanization has a significant influence on urban climate and five of the seven urbanization variables presented significant positive effects on PET except for population density. The most important contribution to changes in thermal comfort is GDP, followed by urban population.

Changes in urban thermal environment caused by urbanization need to be emphasized seriously, especially the cities in the Northeast and North region in China. Our results may help researchers and city planners to well understand PET formation and provide practical guidelines, such as, reasonable planning for an unbalanced regional development of industries for urban planners to improve urban thermal environment.

## ACKNOWLEDGEMENTS

We would like also to provide our great gratitude to the editors and the anonymous reviewers who gave us their insightful comments and suggestions.

### Funding

This research was supported by the Youth Science fund project (41701210) approved by the National Natural Science Foundation of China, Science Development Project of Jilin Province, China (20180418138FG). The funders had no role in study design, data collection and analysis, decision to publish, or preparation of the manuscript.

### Grant Disclosures

The following grant information was disclosed by the authors:
Youth Science fund project (41701210) approved by the National Natural Science Foundation of China.
Science Development Project of Jilin Province, China: 20180418138FG.

### Competing Interests

The authors declare there are no competing interests.

### Author Contributions

- Zhibin Ren conceived and designed the experiments, analyzed the data, contributed reagents/materials/analysis tools, prepared figures and/or tables, authored or reviewed drafts of the paper, approved the final draft.
- Yao Fu performed the experiments, analyzed the data, contributed reagents/materials/-analysis tools, prepared figures and/or tables.
- Yunxia Du conceived and designed the experiments, design the paper.
- Hongbo Zhao conceived and designed the experiments, english Editing.

### Data Availability

Raw data is available in the Supplemental Files. The raw data consists of the historical publicly available meteorological data in 1990, 2005 and 2015 from the National Meteorological Information Center (http://data.cma.cn/) used in our study in order to investigate spatiotemporal trends of urban thermal comfort.

### Supplemental Information

Supplemental information for this article can be found online at http://dx.doi.org/10.7717/peerj.7424#supplemental-information.

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
