# Peer review of "Spatiotemporal patterns of urban thermal environment and comfort across 180 cities in summer under China’s rapid urbanization"

_PeerJ, doi:10.7717/peerj.7424_

## Round 0.1 · original submission · Major Revisions

Two reviewers have provided detailed comments on your manuscript with varying recommendations. Overall the topic of this study is important, however, both reviewers noted the obvious shortcomings.

The review from Reviewer 1 is in their attached PDF

Reviewer 1 ·

Basic reporting

Literature references, sufficient field background/context provided

Experimental design

Research question well defined, relevant & meaningful. It is stated how research fills an identified knowledge gap.

Validity of the findings

Conclusions are well stated, linked to original research question & limited to supporting results.

Additional comments

Major revision

Annotated reviews are not available for download in order to protect the identity of reviewers who chose to remain anonymous.

Reviewer 2 ·

Basic reporting

Please refer to the general comments.

Experimental design

Please refer to the general comments.

Validity of the findings

Please refer to the general comments.

Additional comments

The study has investigated thermal discomfort in 180 cities in China using a variety of datasets for 1990, 2005, and 2015. The topic is interesting and the findings are useful. However, the paper should be revised and the data/methods need to be clarified. There are many writing issues in the paper, and I tried to indicate some of them in the following, but please do a thorough writing check on the manuscript. Please find my comments as follows:

1. Line 64: “exhibit an decreasing” >> “exhibit a decreasing”
2. Line 65: “cities had positive” >> “cities that have positive”
3. Line 68: what does “GDP” stand for? Is it the gross domestic product? If so, the sentence does not read well; “the most important contribution to increasing PET is GDP”. GDP is a monetized measure and does not cause an increase in temperature.
4. Line 90: “become worsen” >> “become worse”
5. Line 112: “critically very necessary” >> “critically necessary”
6. Line 115: “have experienced” >> “has experienced”
7. Line 125: “were still” >> “are still”
8. Lines 126-127: the sentence does not read well. Please rephrase it.
9. Line 127: “an national” >> “a national”
10. Line 160: “was used” >> “were used”
11. Line 161: “Four variables” >> “the four variables”
12. Section 2.3, Calculation of urban thermal comfort: It is not clear how the authors have calculated the Physiological Equivalent Temperature (PET). They only mention that “PET was derived from the human energy balance” (Line 188). Is there an equation to calculate it? Or did the authors use a specific software package? Please explain how PET is derived in details.
13. Line 191: “monthy” >> “monthly”
14. Line 191: “The monthly PET index … was calculated” >> Does it mean that monthly data are used for calculating PET? Please explain the data in more details (temporal resolution that is used, period, variables). You may even consider summarizing the information in a small table.
15. Line 221: Fig.4 is introduced before Figures 2 and 3. The order of figures should be revised.
16. Fig.2 : It is better to revise “air humidity” to “relative humidity”. In addition, the legends do not read well. Please revise them and please include the legend for the green bar (2015) to all three plots.
17. Figures 5 and 6: In these two figures, what does “1990-2005” indicate? Does it show PET1995-PET2005? Based on the conclusions in the text, it seems that it should be 2005-1995, since the authors mention that “95% of cities had positive trend” (Line 241). Please revise the legend in both figures accordingly.
18. Discussion: It is important to discuss about the limitations of this study and their effect on the final conclusion. For instance, it has to be mentioned that the analyses are solely for three different years and the so-called “trends” do not necessarily indicate long-term changes, but they merely show year-to-year differences. Furthermore, it seemed that monthly data was used and then averaged over the season. Therefore, the study does not capture the severity of thermal discomfort neither its peaks, and it only provides a general condition for each city in each year.
Another important discussion to include in the paper is the expected impacts of climate change on the issue. I think the authors may find the following recently published study useful for this matter:
• Sun, Q., Miao, C., Hanel, M., Borthwick, A. G., Duan, Q., Ji, D., & Li, H. (2019). Global heat stress on health, wildfires, and agricultural crops under different levels of climate warming. Environment international, 128, 125-136.

---

## Round 0.2 · accepted · Accept

I am pleased to inform you that your paper has been accepted for publication. You were responsive to most of the comments made by the reviewers.

Reviewer 1 ·

Basic reporting

no comment

Experimental design

no comment

Validity of the findings

no comment

Additional comments

The revisions are very good. I think the manuscript can be accepted for publication.

Reviewer 2 ·

Basic reporting

Pass

Experimental design

Pass

Validity of the findings

Pass

Additional comments

Thank you for point by point response to all the comments. All my suggestions are addressed in the revised paper.